# Molecular Similarity Perception Based on Machine-Learning Models

**DOI:** 10.3390/ijms23116114

**Published:** 2022-05-30

**Authors:** Enrico Gandini, Gilles Marcou, Fanny Bonachera, Alexandre Varnek, Stefano Pieraccini, Maurizio Sironi

**Affiliations:** 1Dipartimento di Chimica, Università degli Studi di Milano, Via Golgi 19, 20133 Milano, Italy; enrico.gandini@unimi.it (E.G.); maurizio.sironi@unimi.it (M.S.); 2Laboratory of Chemoinformatics, UMR 7140, University of Strasbourg, CNRS, 4 Rue Blaise Pascal, 67000 Strasbourg, France; g.marcou@unistra.fr (G.M.); f.bonachera@unistra.fr (F.B.)

**Keywords:** molecular similarity, similarity perception, machine learning, chemical data set

## Abstract

Molecular similarity is an impressively broad topic with many implications in several areas of chemistry. Its roots lie in the paradigm that ‘similar molecules have similar properties’. For this reason, methods for determining molecular similarity find wide application in pharmaceutical companies, e.g., in the context of structure-activity relationships. The similarity evaluation is also used in the field of chemical legislation, specifically in the procedure to judge if a new molecule can obtain the status of orphan drug with the consequent financial benefits. For this procedure, the European Medicines Agency uses experts’ judgments. It is clear that the perception of the similarity depends on the observer, so the development of models to reproduce the human perception is useful. In this paper, we built models using both 2D fingerprints and 3D descriptors, i.e., molecular shape and pharmacophore descriptors. The proposed models were also evaluated by constructing a dataset of pairs of molecules which was submitted to a group of experts for the similarity judgment. The proposed machine-learning models can be useful to reduce or assist human efforts in future evaluations. For this reason, the new molecules dataset and an online tool for molecular similarity estimation have been made freely available.

## 1. Introduction

An orphan drug is a medicinal product used to treat a rare disease that affects only a small number of patients (the actual number of patients depends on the local legislations) [1]. Given the small number of patients affected by the rare disease, and the high costs involved in modern drug discovery programs [2,3], orphan drugs are not an immediately attractive market for pharmaceutical companies.

To encourage pharmaceutical companies to develop orphan drugs, regulatory agencies have brought forward legislation that provides a range of incentives. Such incentives include grants, financial incentives, the possibility of an accelerated review, and market exclusivity. Market exclusivity is arguably the most important incentive: under the EU legislation, a pharmaceutical company that develops an orphan drug for a specific rare disease is given a 10-year period of market exclusivity. During this period, no products that are considered similar to that orphan drug can be accepted or authorized by any European regulatory competent authority. Orphan drugs have less competition than conventional drugs, which encourages pharmaceutical companies to invest in researching novel medicines for rare diseases.

The assessment of similarity between two drugs takes into account three criteria: molecular structure, mechanism of action, and therapeutic indication. Two drugs will be considered diverse if there are significant differences in one or more of the three aforementioned criteria. Thus far, the European Medicines Agency (EMA) has used majority voting on discretional judgments of similarity when assessing new drugs for rare diseases. Similarity is an inherently subjective concept, which depends on individual factors such as gender, age, state of mind, and previous experiences [4,5]. In general, chemical structure information is perceived differently by different individuals [6], but a fair level of consistency can be achieved using a wisdom of crowds approach [7].

Automated procedures that quantitatively evaluate molecular similarity are desirable, and the use of quantitative estimations of molecular similarity is well established in cheminformatics for virtual screening purposes [8,9,10]. Such an algorithm would not replace the current human-based processes used to evaluate applications for orphan drugs authorizations. Instead, it would produce a useful quantitative input to be considered by the human experts evaluating the application. Additionally, such tools could be particularly useful for managing drug design projects; for instance, to decide early to stop the development of a lead because it is too similar to an already marketed drug.

Franco et al. developed Logistic Regression (LogReg) models that calculate the probability that a pair of molecules will be considered similar by a crowd of experts [11,12]. LogReg models relate the opinion of the experts to Tanimoto coefficients calculated on different 2D molecular fingerprints. The similarity between two compounds computed in this way is considered to be a quantitative and objective similarity measure because it is uniquely defined, following a precise algorithmic procedure. These models successfully reproduced human assessments of molecular similarity, both on the data set used to train the LogReg models and on an external test set. Unfortunately, these models were not implemented as public web services and, therefore, are not readily available.

Franco et al. [12] also reported the LogReg models based on 3D molecular fingerprints calculated with the MOE [13] program. These models performed worse than those based on the simpler 2D fingerprints. This was explained in [12] by the fact that too much of the 3D structural information was lost as it was encoded in a 1D bit vector. However, one could also suggest that the 3D MOE descriptors do not capture well enough the information relevant for similarity assessment. Indeed, the ROCS tool of OpenEye [14] considering molecular shape and the spatial orientation of pharmacophoric groups can be more appropriate. Contrary to simple 3D fingerprints, ROCS does not compress 3D molecular information that is held in a 3D numerical tensor, and the similarity measure TanimotoCombo is calculated on a pair of such 3D tensors [15,16,17]. 

When gathering their dataset, Franco et al. [11] presented to the experts only 2D structures; thus, the answers were biased toward perception of 2D similarity, whereas important 3D features like molecular shape, orientation of pharmacophoric features, etc., were completely ignored. Moreover, a test set on which Franco et al. validated their models is not available. Therefore, in this contribution, we describe a new dataset collecting expert opinion about the similarity of pairs of compounds that extend the data reported by Franco et al. [11]. New data were collected using a new online survey for experts to assess molecular similarity on the new pairs of compounds. In this survey, the experts were provided with both 2D structures and an optimal alignment of 3D structures of the compounds. The survey analysis reveals the importance of 3D information on the decision of human experts when comparing two chemical structures. 

We used the survey results to produce new LogReg models predicting for a pair of compounds the likeliness to be considered as similar or dissimilar by a panel of experts. These models are publicly available on our WEB site (https://chematlas.chimie.unistra.fr/ReadySim/ (accessed on 26 May 2022)). We suggest using these publicly available models in agencies, such as EMA, in order to focus attention on those cases where the similarity is predicted to be debatable in a panel of experts. Such tools will also be useful for pharmaceutical companies and drug designers to take objective decisions regarding the development of a lead suitable for receiving the orphan drug status. Our WEB-based tool for similarity prediction takes as input a pair of molecules, which can be either drawn by the user into a graphical interface or submitted in SMILE format, and gives as output the probability that the pair will be considered similar by a panel of experts.

## 2. Results and Discussion

### 2.1. Rational Selection of the Dataset and Human Experts Similarity Assessment 

The new dataset was created in order to consider a wide range of molecular similarity scenarios, including molecular pairs (MP) that are rather difficult to subject to human analysis. Selection of molecular pairs was performed on the basis of both 2D and 3D similarity measures. The former was approximated by Tanimoto coefficient calculated with CDK Extended fingerprints ($t_{\mathrm{XT}}$) providing with the best LogReg model on the dataset by Franco et al. [11]. The latter was assessed as a TanimotoCombo metric ($t_{\mathrm{CS}}$) calculated with the ROCS tool of OpenEye [14]. 

A set of 9000 bioactive MP was randomly selected from the ChEMBL 27 database [18] using subsets of ligands against three well-known biological targets HERG [19], 5HT2B [20], and CYP2D6 [21], see Section 3 for the details. To classify molecular pairs as either similar or dissimilar in 2D and in 3D, we used an approach based on a similarity threshold with a small buffer region, similar to the one described by Ehrman et al. [22]. We classified a molecular pair as similar in 2D if $t_{\mathrm{XT}}$ ≥ 0.7, and as similar in 3D if $t_{\mathrm{CS}}$ ≥ 1.4. Such thresholds are popularly used for the two similarity measures [23,24,25]. To avoid an extreme sensitivity to small molecular differences around the thresholds [17], we used a 0.05 and a 0.1 buffer region for $t_{\mathrm{XT}}$ and $t_{\mathrm{CS}}$, respectively. Therefore, we classified a molecular pair as dissimilar in 2D if $t_{\mathrm{XT}}$ ≤ 0.65, and as dissimilar in 3D if $t_{\mathrm{CS}}$ ≤ 1.3. Extracted data were then divided into four subsets: two subsets in which the 2D and 3D similarity measures agreed on the similarity (*sim2D*,*sim3D*) or dissimilarity (*dis2D*,*dis3D*) of the MP, and two subsets in which the calculated similarity measures were diametrically opposite for a given MP: *sim2D*,*dis3D* and *sim3D*,*dis2D* (see Section 3 for further details). We selected 25 MP for each of the four subsets, and challenged experts to label them as similar or dissimilar through the online survey. The web application that we developed for this task allowed the survey users to freely inspect the 3D representations of each MP, while at the same time observing the 2D molecular graphs. The 3D structures corresponded to the best alignment found with ROCS. In contrast to the survey reported by Franco et al. [11,12], the experts were free to resort to both 2D and 3D representations to make their decision. 

The survey was completed by 418 users: 61.5% of them were professors or researchers, 7.4% were postdocs, and 16.7% were PhD students. The remaining 14.4% of the users reported to not possess any of the aforementioned academic titles. A total of 2090 MP assessments were collected: each of the 418 users assessed five randomly selected MP. The number of assessments per MP varied from 11 to 30. On average, each MP received 21 assessments. The MP in the four calculated similarity subsets received a comparable number of similarity assessments.

The $t_{XT}$ and $t_{CS}$ measures of the MP in each of the four subsets are given in Figure 1, with 2D structures of some representative MP. The *dis2D*,*dis3D* subset includes two main types of MP. Some pairs are dissimilar in 2D and in 3D because they are of very different sizes. This subset also includes MP that are of comparable sizes, but with different chemical functionalities and shapes. The *dis2D*,*sim3D* subset includes MP with similar size, shape, and relative orientation of functional groups, but with somewhat different chemical functionalities. The *sim2D*,*dis3D* subset includes MP whose 2D graph is fairly similar; they are of similar size, and have similar chemical functionalities placed in different positions of the basic scaffold. Their diversity is more apparent when observing their 3D representations. Finally, the *sim2D*,*sim3D* subset includes molecules that are highly similar in 2D and in 3D; they are of similar size, with similar scaffolds, similar chemical functionalities in similar positions.

The selected subsets are in excellent agreement with the similarity assessments by survey users (Figure 2). Molecular pairs in the *sim2D*,*sim3D* subset are considered similar by a high percentage of users (81.7% on average). On the other hand, users considered molecular pairs belonging to the *dis2D*,*dis3D* subset to be dissimilar (92.0% on average). As we expected, users did not agree very strongly on the similarity of molecules in the *sim2D*,*dis3D* and *dis2D*,*sim3D* subsets (55.5 and 50.7% respectively). 

It should be noted that the $t_{CS}$–$t_{XT}$ plot for the Franco et al. data set (Figure 1, bottom) contrasts with that for the dataset collected in this work. Indeed, most of the data points are situated near the diagonal of the plot corresponding to the presence of the *sim2D*,*sim3D* and *dis2D*,*dis3D* subsets only. The presence in our data set of more problematic *sim2D*,*dis3D* and *dis2D*,*sim3D* subsets makes the similarity prediction task more challenging.

### 2.2. Building and Validation of the Models

Similar to Franco et al., we developed machine-learning models aiming to predict human assessment of similarity for molecular pairs. One- and two-feature Logistic Regression models were built on the collected data using Equations (2) and (4), respectively (see Section 3). The Franco data set was used for the models’ validation. For the sake of comparison, the LogReg models were also developed on the Franco data set then validated on the data collected in this work. 

Table 1 resumes the performance of the models built on the collected data set. One can see that at the fitting stage, statistical parameters are rather modest: the model involving 2D similarity calculated with CDK Extended fingerprints ($t_{XT}$-model) performs much better than that built on 3D TanimotoCombo similarity ($t_{CS}$-model). Thus, the number of correctly predicted molecular pairs (out of 100, N_correct_) is 81 and 70 for $t_{XT}$- and $t_{CS}$-models, respectively. The model involving both $t_{XT}$ and $t_{CS}$ variables performd slightly better (N_correct_ = 84), but still not perfectly. On the other hand, at the validation stage, all models demonstrated very good performance on the Franco set; the number of correctly predicted molecular pairs (out of 100) was 92 for both single-feature models and 95 for the double-feature models (Table 1).

As illustrated in Figure 3, at the training stage, the single-featured models correctly predicted 100% of the molecular pairs in the *sim2D*,*sim3D* and *dis2D*,*dis3D* subsets. All the prediction errors occurred in the *sim2D*,*dis3D* and *dis2D*,*sim3D* subsets. On the *dis2D*,*sim3D* subset, both the $t_{XT}$- and $t_{CS}$-models performed poorly: 52 and 48% correct predictions each. On the other hand, the $t_{XT}$-model better fitted the *sim2D*,*dis3D* subset (72% correct predictions), whereas the $t_{CS}$-model performed very poorly on the same subset (32%). We hypothesize that this difference in 2D and 3D model expressivity on this subset may be explained by the fact that humans, when in doubt, tend to consider only the 2D molecular graphs, whose similarity is well represented by the $t_{XT}$ values. This behavior may explain why the $t_{XT}$ values are a better predictor than $t_{CS}$ for difficult cases of the similarity prediction task.

In contrast to the models built on the collected set, those trained on the Franco set demonstrated a reasonable performance at the training stage, but performed poorly at the validation stage (see Table 2). Indeed, being applied to the collected set, the single-feature $t_{XT}$- and $t_{CS}$-models correctly predicted 81 and 69 molecular pairs, respectively. Using a double-feature equation employing both $t_{XT}$ and $t_{CS}$ did not improve the performances compared to the $t_{XT}$-model alone.

Using Equation (4) and the single-feature model for the collected set, we identified the thresholds $t_{XT}\geq 0.73$ ($t_{XT}\leq 0.42$) and $t_{CS}\geq 1.62$ ($t_{CS}\leq 0.89$) corresponding to 95% of expert opinion being that a given molecular pair is similar (or dissimilar).

## 3. Materials and Methods

### 3.1. The Data Set Used for Human Assessments

We created a new dataset of human assessments of molecular similarity to replace the missing test set from the original Franco et al. work. We changed the design of this dataset in order to investigate new situations. This new dataset had to include pairs of compounds that could display a significant 2D dissimilarity but be able to participate in equivalent 3D interactions. For this reason, we decided to select compounds binding to proteins with well-defined binding sites. Since the new dataset needed also pairs of compounds sharing a certain degree of 2D similarity but admittedly different 3D pattern of interaction, we investigated ligands of cytochromes. Finally, we queried the ChEMBL 27 database [18,26,27] for molecules that targeted three well-known biological targets: HERG [28], 5HT2B [20], and CYP2D6 [21]. We included only compounds for which an inhibition constant was measured, using the pChEMBL values [27]. We selected 1307 compounds that targeted HERG, 1299 compounds that targeted 5HT2B, and 155 compounds that targeted CYP2D6. We used InChiKey [29,30] to remove duplicate compounds and visually verified the resulting dataset. This protocol ensured that we were able to recover enough compounds.

We applied the 2D protocol to all compounds, calculating Tanimoto CDK Extended between each unique MP. We then applied the OpenEye 3D protocol. Several molecules did not pass the Omega conformer generation step, and were discarded. We randomly selected 3000 molecular pairs for each target, and performed ROCS alignment and scoring between all conformers of each of the 9000 total MP. This dataset has been divided in 4 subsets: pairs that are similar in 2D and in 3D (*sim2D*,*sim3D*), pairs that are similar in 2D and dissimilar in 3D (*sim2D*,*dis3D*), pairs that are dissimilar in 2D and similar in 3D (*dis2D*,*sim3D*), and pairs that are dissimilar in 2D and dissimilar in 3D (*dis2D*,*dis3D*). We finally selected 25 pairs from each set. We thus obtained a data set with 100 MP, containing 25 pairs from each similarity subset. 

We developed a web-survey application. We used Voilà [31], a tool for converting Jupyter notebooks [32] in standalone web applications. The web application was served on the Heroku Cloud Application Platform [33]. We sent invitations to take part to the survey to 69 chemistry departments and institutions worldwide. The survey was available on Heroku from 14 April 2021 to 28 June 2021. The results were automatically stored by the web application on a private PostgreSQL [34] database available through Heroku.

The web application would present 5 randomly selected MP to each survey users. For each MP, users were presented the static 2D graph pictures of the molecules (already aligned to the Maximum Common Subgraph with RDKit). Users were also shown 3D interactive molecular representations: the best ROCS alignments of *Omega* conformers were selected, and presented to users with molecular visualization tool NGLview [35,36]. Users were free to interact with the NGLview molecular representations, and could return to the initial well-centered representations by clicking a “Reset 3D Views” button.

Users had to express a similarity assessment for each of the 5 MP that were presented to them. The application did not allow users to proceed in the survey without answering. Users could not go back and change similarity assessments of previous molecules. After the 5 similarity assessments were expressed, users were asked about their academic qualification. The application stored only the answers of users who completed the survey. 

### 3.2. The Franco et al. Training Set

The first set of models that we developed is based on the original training set created and kindly made available by Franco et al. [11]. It consists of 100 MP downloaded from DrugBank 3.0 [37], and selected to cover the widest and most uniform spread of Tanimoto values computed on ECFP4 fingerprints [38]. The 100 MP were evaluated by 143 experts from international regulatory authorities. The experts were asked to evaluate whether each molecular pair was composed by similar (*Yes*) or dissimilar (*No*) molecules. The authors then calculated the percentage ($p_{xp}$) of experts that considered each MP to be similar. They labeled as similar the MP that were considered similar by ≥50% of experts. 

Franco et al. used a test set containing confidential information provided by EMA’s Committee for Medicinal Products for Human Use (CHMP). We asked CHMP to provide us confidential access to the original test set. Our request was kindly approved, but at the time of writing we did not receive the dataset.

### 3.3. The 2D Protocol

The protocol for building 2D similarity prediction models involved preprocessing of original SMILES using RDKit [39] and MolVS [40] for standardization. Counterions were removed, and the remaining species neutralized. We then visually inspected all the molecules. After preprocessing, we computed all 2D fingerprints available in RDKit and CDK [41,42], and calculated Tanimoto coefficients on each MP, with each 2D fingerprint.

### 3.4. The OpenEye 3D Protocol

The other 3D protocol was based on OpenEye software. The first step of the protocol was SMILES preprocessing with *Filter* command (included in OMEGA [43] software). The *Filter* tool was set to standardize the structures, but not to discard any of them. We used the *Omega* with the classic algorithm to generate up to 200 conformers for each molecule. Conformers generated by *Omega* are ready to use, since *Omega* was developed to sample the conformational space around solid-state structures of drug-like molecules [44,45]. For each MP, we use ROCS to perform all possible conformer alignments, and to calculate similarity scores for each alignment. For each MP, we kept the largest TanimotoCombo score value corresponding to the best alignment. In contrast to other similarity measures, the TanimotoCombo takes values between 0 and 2. We term this value, the “ComboScore ROC Similarity”, $t_{CS}$.

### 3.5. Training of Similarity Prediction Models

A Logistic Regression (LogReg) model is built according to Equation (1) for a logistic transform of the percentage of expert opinions $p_{xp}$. Two types of similarity prediction models were considered “single-feature models” (Equation (2)) and “double-feature models” in the present work (Equation (3)). The input feature in Equation (2) is either Tanimoto coefficient calculated on CDK Extended fingerprints (Tanimoto CDK Extended, $t_{XT}$) or TanimotoCombo, $t_{CS}$. Both $t_{XT}$ and $t_{CS}$ are used as an input in Equation (3).

The predicted percentage of experts with the opinion that a pair is similar, ${\hat{p}}_{xp}$, is deduced from the predicted logit value, $\hat{y}$, Equation (4)
(1)$$y=\log\frac{p_{xp}}{1-p_{xp}}$$
(2)$$y=\omega_{0}+\omega_{1}t_{i}$$
(3)$$y=\omega_{0}+\omega_{1}t_{XT}+\omega_{2}t_{CS}$$
(4)$${\hat{p}}_{xp}=\frac{e^{\hat{y}}}{1+e^{\hat{y}}}$$

The LogReg models were built using scikit-learn [46] using L1 regularization with the default value for the regularization parameter (λ=1). The coefficients $\omega_{0}$, $\omega_{1}$ and $\omega_{2}$ in Equations (2) and (3), trained on collected data set, are resumed in Table 3. The calculation of $t_{CS}$ requires an OpenEye license, so only Equation (2) can be used if the license is missing. 

### 3.6. Model Performance Evaluation

We evaluated the similarity prediction models using a variety of performance metrics for classification problems [47]. We focused on the number of samples that a model correctly classifies (N_correct_) and the Area Under the Receiver Operating Characteristic curve (ROC AUC).

### 3.7. Model Implementation

The single-feature $t_{XT}$ model is implemented in our server and is publicly available online at https://chematlas.chimie.unistra.fr/ReadySim/ (accessed on 26 May 2022) (Figure 4). This service guarantees that the chemical structures are properly standardized and that the similarities are computed as described in this article.

## 4. Conclusions

In this paper, we re-investigated the work of Franco et al., aiming to model the decision of a panel of experts concerning the similarity of pairs of organic drug-like molecules. In this context, we have gathered a new dataset comprising pairs of compounds that are likely to be perceived as similar in 2D while being dissimilar in 3D, and vice versa. The datasets are publicly available (Zenodo: 10.5281/zenodo.6472293).

Building and testing models, we observed that predicted 2D similarity is the dominant factor corelating with the perception of similarity by the experts. However, the addition of 3D similarity concepts improves the model. This can originate from a bias of the experts to favor 2D information over 3D one when in doubt.

Models built on the new dataset can be used to calculate the expected outcome of majority votes on molecular similarity. The models can be used as an additional tool by agencies tasked with evaluating molecules for the status of orphan drug: since such agencies rely on majority voting as their main decision-making tool, comparing their judgement with the output of a computational model trained to reproduce expert assessments on such a complex data set can be useful. According to the developed models, similarity thresholds $t_{\mathrm{XT}}\geq 0.73$ ($t_{\mathrm{XT}}\leq 0.42$) and $t_{\mathrm{CS}}\geq 1.62$ ($t_{\mathrm{CS}}\leq 0.89$) correspond to a 95% probability that a panel of expert would consider two molecules as similar (dissimilar). In this case, agencies would not really need to consult a panel of experts to make a decision. Instead, they can use our web service (https://chematlas.chimie.unistra.fr/ReadySim (accessed on 26 May 2022) to compute the predicted similarity. The models can also be used as a tool by pharmaceutical companies to perform a preliminary screening of molecules that may be suitable to receive the orphan drug status.

## Figures and Tables

**Figure 1 ijms-23-06114-f001:**
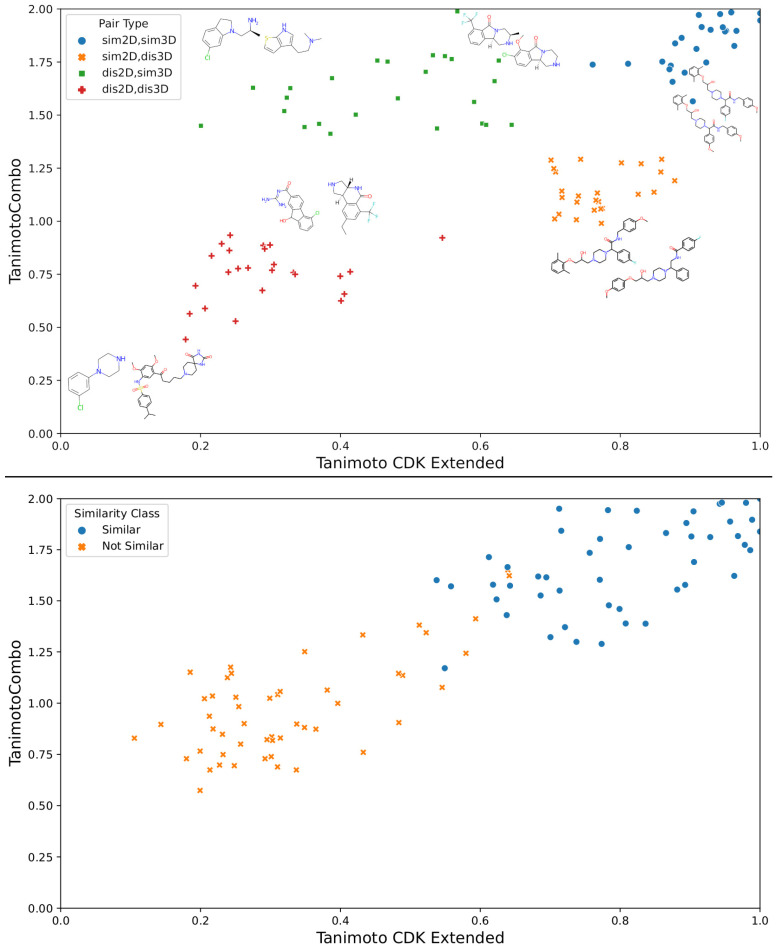
2D (*t_XT_*)/3D (*t_CS_*) similarity plots of MP (**top**) included in the survey and (**bottom**) studied by Franco et al. [11]; 2D structures of some representative molecular pairs are shown (**top**).

**Figure 2 ijms-23-06114-f002:**
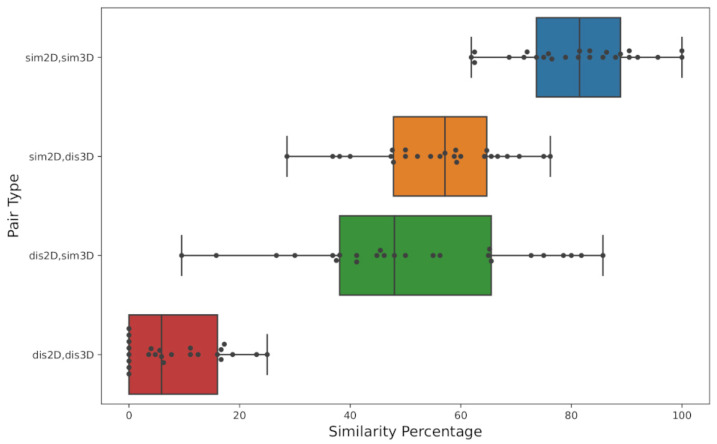
Distribution of molecular pairs (MP) according to human assessed similarity (horizontal axis) in each selected subset.

**Figure 3 ijms-23-06114-f003:**
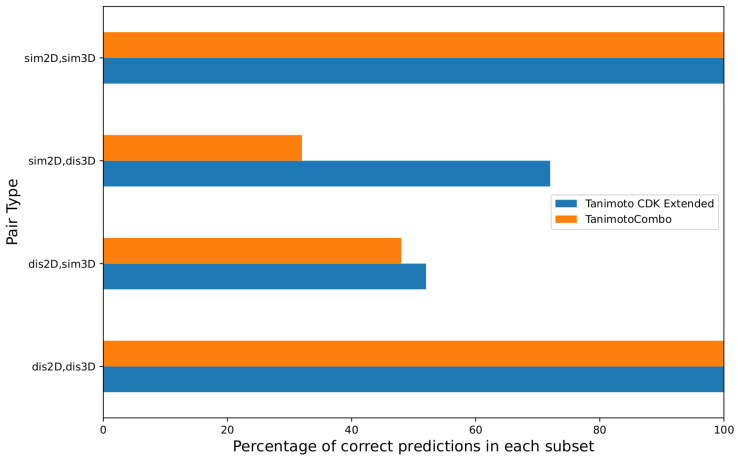
Percentage of predictions by the Tanimoto CDK Extended (*t_XT_*) and TanimotoCombo (*t_CS_*) models in the four calculated similarity subsets of the collected set. Both models correctly predicted 100% of the molecular pairs in the *sim2D*,*sim3D* and *dis2D*,*dis3D* subsets. All prediction errors occurred in the *sim2D*,*dis3D* and *dis2D*,*sim3D* subsets.

**Figure 4 ijms-23-06114-f004:**
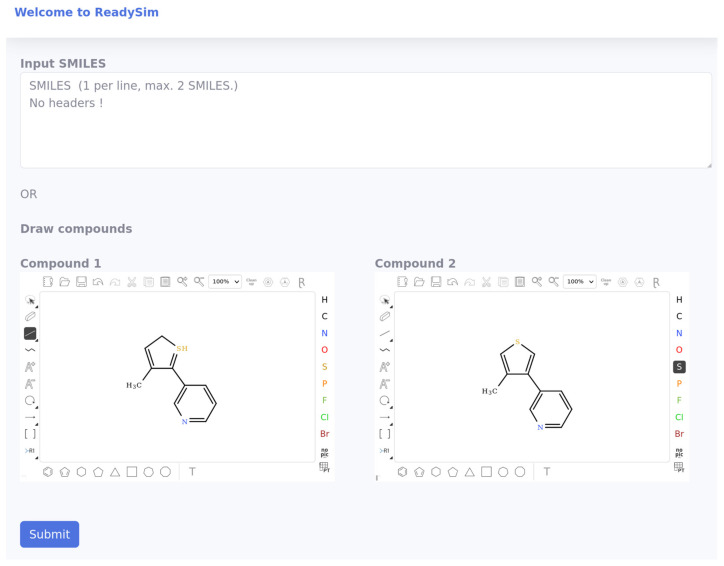
The web service ReadySim. The user can type in a molecular pair either as two SMILES strings or using the chemical sketchers below. The server standardizes the structures, computes the similarity and transforms it back as a probability that the pair will be considered as similar by a panel of experts.

**Table 1 ijms-23-06114-t001:** Models built on the training set collected here and validated on the Franco data set using single-feature (Equation (2)) and double-feature (Equation (4)) logistic regression (LogReg) models. The sizes of the collected and Franco sets are equal to 100 molecular pairs.

Model Type	Variables	Fit		Validation	
		N_correct_	ROCAUC	N_correct_	ROCAUC
single-feature	*t_XT_*	81	0.920	92	0.988
	*t_CS_*	70	0.845	92	0.970
double-feature	*t_XT_*, *t_CS_*	84	0.924	95	0.988

**Table 2 ijms-23-06114-t002:** Models built on the Franco training set and validated on the dataset collected here ^a^.

Model Type	Variables	Fit		Validation	
		N_correct_	ROCAUC	N_correct_	ROCAUC
single-feature	*t_XT_*	93	0.988	81	0.920
	*t_CS_*	91	0.970	69	0.845
double-feature	*t_XT_*, *t_CS_*	95	0.988	81	0.916

^a^ see caption for Table 1.

**Table 3 ijms-23-06114-t003:** Coefficients of Equations (2) and (3) for the models built on the collected set.

	*ω* _0_	*ω* _1_	*ω* _2_
Equation (2), *t_XT_*	−4.860	8.449	-
Equation (2), *t_CS_*	−4.464	3.554	-
Equation (3)	−5.605	5.214	2.009

## Data Availability

Datasets are available under MIT license on Zenodo: 10.5281/zenodo.6472293. Software material is available on the git: https://github.com/enricogandini/paper_similarity_prediction.git (accessed on 26 May 2022).

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
