# Peer review of "Molecular Similarity Perception Based on Machine-Learning Models"

_ijms, 2022, doi:10.3390/ijms23116114_

Round 1

Reviewer 1 Report

Numerous citation errors in your manuscript that are not allowable until fixed for proper review. Specifically see these lines: 146, 186, 307, 308, and 334, which contain these errors:

  • "survey users (Error! Reference source not found.). Molecular pairs in the sim2D,sim3D" --line 146
  • "As illustrated on Error! Reference source not found., at the training stage the single-" --line 186
  • "?2 in equations ( 2 ) and Error! Reference source not found. trained on collected here" --line 307
  • "data set are resumed in Table 3. Coefficients of the equations ( 2 ) and Error! Reference" --line 308

Citation... "are publicly available (Zenodo: 10.5281/zenodo.6472293)". --line 334 

Key Franco publications noteworthy of inclusion within the introduction on the use of similarity modeling for chemical space deconvolution (methods leading up to his current uses of Logistic Regression in ML and AI approaches with fingerprint modeling circa 2013)... historically relevant.
(1) Pérez-Villanueva J, Medina-Franco JL, Caulfield TR, Hernández-Campos A, Hernández-Luis F, Yépez-Mulia L, Castillo R. Eur J Med Chem. 2011 Aug;46(8):3499-508. doi: 10.1016/j.ejmech.2011.05.016. Epub 2011 May 13. PMID: 21621311
(2) Medina-Franco JL, Caulfield T. Drug Discov Today. 2011 May;16(9-10):418-25. doi: 10.1016/j.drudis.2011.02.003. Epub 2011 Feb 16. PMID: 21315180 
(3) Integrating virtual screening and combinatorial chemistry for accelerated drug discovery. López-Vallejo F, Caulfield T, Martínez-Mayorga K, Giulianotti MA, Nefzi A, Houghten RA, Medina-Franco JL. Comb Chem High Throughput Screen. 2011 Jul;14(6):475-87. doi: 10.2174/138620711795767866. PMID: 21521151 

Not extremely rigorously presented, but offers alternative interpretations on a control dataset previously predicated upon for Franco technique, which here is elaborated and given more complexity.
Human survey data was included to examine perception on 2D structures vs reality of 3D data (fingerprints using LogReg models)
Overall presents a new use of 3D fingerprinting with overcoming the obstacles of majority voting algorithms in the 2D methods, which can present as an all/none, rather than the graduated levels that's likely. Somewhat useful tool for prescreening orphan drugs as webserver is given as utility of this project which will have be helpful.

Importance of the topic might be based on predication of the importance of expert understanding of 2D depictions vs comprehension of 3D... But there seems to be interest amongst cheminformatics experts on building on this kind of data, which perhaps can offer AI better insights in the future...

Would recommend major revisions to fix those egregious typographical errors, include key historical references, and clarify the conclusion (bit muddy reading).

Regards,

Author Response

Please find enclosed the answer to reviewer'w comments:

Numerous citation errors in your manuscript that are not allowable until fixed for proper review. Specifically see these lines: 146, 186, 307, 308, and 334, which contain these errors:

  • "survey users (Error! Reference source not found.). Molecular pairs in the sim2D,sim3D" --line 146
  • "As illustrated on Error! Reference source not found., at the training stage the single-" --line 186
  • "?2 in equations ( 2 ) and Error! Reference source not found. trained on collected here" --line 307
  • "data set are resumed in Table 3. Coefficients of the equations ( 2 ) and Error! Reference" --line 308

Citation... "are publicly available (Zenodo: 10.5281/zenodo.6472293)". --line 334 

Broken links and typos have been fixed in the revised manuscript.

Key Franco publications noteworthy of inclusion within the introduction on the use of similarity modeling for chemical space deconvolution (methods leading up to his current uses of Logistic Regression in ML and AI approaches with fingerprint modeling circa 2013)... historically relevant.
(1) Pérez-Villanueva J, Medina-Franco JL, Caulfield TR, Hernández-Campos A, Hernández-Luis F, Yépez-Mulia L, Castillo R. Eur J Med Chem. 2011 Aug;46(8):3499-508. doi: 10.1016/j.ejmech.2011.05.016. Epub 2011 May 13. PMID: 21621311
(2) Medina-Franco JL, Caulfield T. Drug Discov Today. 2011 May;16(9-10):418-25. doi: 10.1016/j.drudis.2011.02.003. Epub 2011 Feb 16. PMID: 21315180 
(3) Integrating virtual screening and combinatorial chemistry for accelerated drug discovery. López-Vallejo F, Caulfield T, Martínez-Mayorga K, Giulianotti MA, Nefzi A, Houghten RA, Medina-Franco JL. Comb Chem High Throughput Screen. 2011 Jul;14(6):475-87. doi: 10.2174/138620711795767866. PMID: 21521151 

Missing citations have been added in the introduction section.

Would recommend major revisions to fix those egregious typographical errors, include key historical references, and clarify the conclusion (bit muddy reading).

The conclusion section has been modified to make the reading more fluid.

The revised manuscript is atached with modification higlighted in yellow.

Reviewer 2 Report

Dear authors,

The article "Molecular similarity perception based on machine-learning models" describes a web service that predicts two molecules' similarity using regression models. The authors claim that this database will be helpful for pharmaceutical companies and drug designers. The article is helpful; the authors well describe the methods through which they built the web service. The model used for predictions has good statistical parameters, and the results are encouraging. However, I have some comments for the article that should be included before it is published:

Abstract:

  1. The abstract section is unclear, and I suggest that the abstract be re-written.
  2. This part is not relevant to the abstract section and should be put in the introduction. “The developed models as well as created dataset are publicly available on ZENODO (doi: 10.5281/zenodo.6472293).”

Introduction:

  1. The article will benefit if the ReadySim web service is better detailed in this section. ReadySim does not have a description. I agree that the web service is user-friendly. Users can enter SMILES codes, drawing compounds, and open structures, but the input accepted format is not specified. The web service is described in the materials and methods section, but I recommend a small description in the introduction as well.

Results and discussion:

  1. There are some missing references in this part, with the message: “(Error! Reference source not found." The message appears in rows 146 and186.
  2. The figures are too small and hard to analyse. Please provide better figures.
  3. Figure 1 appears two times.
  4. Figure 2 does not have a title on the X-axis.
  5. This section is a little ambiguous. Paragraphs in the "results and discussion" section may belong in the "materials and methods" section, and vice versa. More clarity is necessary.

Materials and Methods:

  1. There are some missing references in this part, with the message: “(Error! Reference source not found." The message appears in rows 307, 308, 312 and Table 3.
  2. Figure 4 is too small.

Overall, the work is intriguing, and the findings are promising enough to serve as a possible starting point for drug design research. However, the text is difficult to comprehend, and the paper might benefit from more clarity and explanations in the “Results and Discussion” and “Materials and Methods” parts. Also, extensive editing of the English language and style is required

Author Response

Please find enclosed the answer to reviewer's comments:

  1. The abstract section is unclear, and I suggest that the abstract be re-written.
  2. This part is not relevant to the abstract section and should be put in the introduction. “The developed models as well as created dataset are publicly available on ZENODO (doi: 10.5281/zenodo.6472293).”

The abstract has been completely rewritten and the links have been removed.

Introduction:

  1. The article will benefit if the ReadySim web service is better detailed in this section. ReadySim does not have a description. I agree that the web service is user-friendly. Users can enter SMILES codes, drawing compounds, and open structures, but the input accepted format is not specified. The web service is described in the materials and methods section, but I recommend a small description in the introduction as well.

A brief introduction to the RedySim web service has been written in the introduction, specifying the input mode.

Results and discussion:

  1. There are some missing references in this part, with the message: “(Error! Reference source not found." The message appears in rows 146 and186.

Typos and broken links have been fixed

  1. The figures are too small and hard to analyse. Please provide better figures.

All the figures have been replaced with higher definition ones.

  1. Figure 1 appears two times.

Duplication has been removed

  1. Figure 2 does not have a title on the X-axis.

The missing X-axis label has been added

  1. This section is a little ambiguous. Paragraphs in the "results and discussion" section may belong in the "materials and methods" section, and vice versa. More clarity is necessary.

The reviewer comments was not very precise on this point. We believe that after our corrections, the organization of the manuscript has improved enough

Materials and Methods:

  1. There are some missing references in this part, with the message: “(Error! Reference source not found." The message appears in rows 307, 308, 312 and Table 3.

Missing reference has been fixed

  1. Figure 4 is too small.

A larger version of figure has been used

All the modifications in the paper have been highlighted in yellow in the attached file.

Round 2

Reviewer 1 Report

Improvements matched this reviewer's requests. Thank you.